# EMSIG: Uncovering Factors Influencing COVID-19 Vaccination Across Different Subgroups Characterized by Embedding-Based Spatial Information Gain

**DOI:** 10.3390/vaccines12111253

**Published:** 2024-11-04

**Authors:** Zongliang Yue, Nicholas P. McCormick, Oluchukwu M. Ezeala, Spencer H. Durham, Salisa C. Westrick

**Affiliations:** 1Department of Health Outcomes Research and Policy, Harrison College of Pharmacy, Auburn University, Auburn, AL 36849, USA; zzy0065@auburn.edu (Z.Y.); nzm0066@auburn.edu (N.P.M.); ome0004@auburn.edu (O.M.E.); 2Department of Pharmacy Practice, Harrison College of Pharmacy, Auburn University, Auburn, AL 36849, USA; durhash@auburn.edu

**Keywords:** COVID-19, vaccination, hesitancy, uptake, factor, subgroup, embedding, spatial information gain, machine learning

## Abstract

**Background/Objectives:** COVID-19 and its variants continue to pose significant threats to public health, with considerable uncertainty surrounding their impact. As of September 2024, the total number of deaths reached 8.8 million worldwide. Vaccination remains the most effective strategy for preventing COVID-19. However, vaccination rates in the Deep South, U.S., are notably lower than the national average due to various factors. **Methods:** To address this challenge, we developed the Embedding-based Spatial Information Gain (EMSIG) method, an innovative tool using machine learning techniques for subgroup modeling. EMSIG helps identify subgroups where participants share similar perceptions but exhibit high variance in COVID-19 vaccine doses. It introduces spatial information gain (SIG) to screen regions of interest (ROI) subgroups and reveals their specific concerns. **Results:** We analyzed survey data from 1020 participants in Alabama. EMSIG identified 16 factors encompassing COVID-19 hesitancy and trust in medical doctors, pharmacists, and public health authorities and revealed four distinct ROI subgroups. The five factors, including COVID-19 perceived detriment, fear, skepticism, side effects related to COVID-19, and communication with pharmacists, were commonly shared across at least three subgroups. A subgroup primarily composed of Democrats with a high flu-shot rate expressed concerns about pharmacist communication, government fairness, and responsibility. Another subgroup, characterized by older, white Republicans with a relatively low flu-shot rate, expressed concerns about doctor trust and the intelligence of public health authorities. **Conclusions**: EMSIG enhances our understanding of specific concerns across different demographics, characterizes these demographics, and informs targeted interventions to increase vaccination uptake and ensure equitable prevention strategies.

## 1. Introduction

COVID-19 vaccination hesitancy is one of the greatest challenges that must be addressed to enhance community protection against the virus and its potential variants, especially in anticipation of a future pandemic. Coronavirus Disease 2019 (COVID-19) is among the top 10 causes of death, having caused 8.8 million deaths worldwide, according to public data reports [1]. Vaccination remains the most effective strategy for preventing COVID-19 [2,3]. Despite the safety and effectiveness of COVID-19 vaccinations in reducing hospitalizations and mortality [4], its uptake in the Deep South, United States (U.S.), which includes Alabama, Georgia, Louisiana, Mississippi, and South Carolina, remains consistently over 10% lower than the national average [5], sparking concern in this geographic region. In 2022, the COVID-19 vaccination rate for individuals aged 12 years and older was 63% in the Deep South compared to 75% nationwide [5]. Similarly, the COVID-19 booster uptake rate in the Deep South was 40%, compared to 51% nationwide [5]. Based on population studies in the Deep South, disparities in COVID-19 vaccination and booster rates may be influenced by factors such as age [5] and political party affiliation [6,7,8,9,10,11,12]. The booster uptake rate was higher among individuals aged 65 and above compared to adults aged 18 to 64 and youth aged 12 to 17 [5]. Low booster uptake is further exacerbated by the tendency of Republicans in the U.S. to be less likely to hold accurate beliefs about vaccines compared to Democrats [6], especially in the Deep South states, which are predominantly Republican. Political ideology is a key factor in explaining why some people hesitate to receive a vaccine, according to research on COVID-19 vaccine hesitancy [10,11]. Additionally, individuals with limited knowledge of COVID-19 tend to have anti-vaccine policy preferences and overestimate their understanding of vaccines compared to experts [13,14]. Conversely, individuals with a good understanding of COVID-19 are more likely to have a positive attitude toward vaccination [15,16]. Therefore, the lower vaccine uptake rate in the Deep South may be attributed to party affiliation and limited knowledge of COVID-19, leading to vaccine hesitancy.

Several specialized scales have been developed to identify and quantify vaccine hesitancy as well as the explanatory factors behind vaccine hesitancy. These scales are crucial for uncovering the specific factors contributing to vaccine hesitancy in the Deep South. Vaccine hesitancy can be directly linked to concerns about the vaccine itself [17] or trust in medical providers, trust in pharmacists [18], and trust in public health authorities [19]. The multidimensional COVID-19 Vaccine Hesitancy Scale (CoVaH) [17] introduced a three-factor structured scale—Skepticism, Risk, and Fear—capturing meaningful constructs related to COVID-19 vaccination uptake. This scale demonstrated good overall specificity and sensitivity in differentiating unvaccinated individuals from vaccinated ones. The Trust in Doctors in General (T-DiG) Scale [20] introduced seven subscales—Communication competency, Fidelity, Systems trust, Confidentiality, Fairness, Stigma-based discrimination, and Global trust—with sound psychometric properties. This scale is useful for researchers evaluating trust-related interventions or conducting studies where trust in medical doctors, such as physicians, is a significant construct or main outcome. The Trust in Community Pharmacists (TRUST-Ph) [18] measure introduced three dimensions—Benevolence, Technical Competence, and Communication—to evaluate patient trust in community pharmacists, demonstrating relatively high validity and reliability. This scale can potentially be used to measure patient-reported perceptions of community pharmacists. The Trust in Public Health Authorities (TiPHA) scale [19] introduced a two-factor—Beneficence and Competence—structural scale, which facilitated the development of targeted interventions to address low trust and improve compliance with public health authorities during routine and emergency public health activities. It is critical to comprehensively include multiple factors that may influence vaccination intention. These studies [15,21,22,23] evaluated the effect of perceived vaccination efficacy, effectiveness, trust in corporations, trust in health authorities, and trust in doctors on vaccination intention.

Data-driven analysis and subgroup identification are valuable for informing policies and messages to target different vulnerable groups within a population [24]. Additionally, subgrouping is critical for identifying diverse barriers and informing targeted interventions to address the concerns of vulnerable groups, thereby increasing vaccination rates. The challenge arises from the fact that no single intervention is likely to address vaccine hesitancy comprehensively, and identifying the diverse concerns and barriers to vaccination uptake at a subpopulation level is essential, particularly in groups with lower vaccination rates [25]. The unsupervised learning approach provides an unbiased method to address these challenges [24,26]. For instance, K-Means clustering has been employed to characterize participants based on demographics, knowledge, perceptions, risks, and other factors. This approach helps identify specific archetypes or subgroups within the cohort, determining if certain groups are more likely to take the COVID-19 vaccine. However, several limitations exist in clustering algorithms, including their limited predictive power for vaccination uptake or doses, the inclusion of incoherent participants in the clustering process, and the absence of an explicit cutoff for subgroup formation. Moreover, long-standing beliefs that predate the pandemic contribute to distrust in government [27] and foster reluctance to receive vaccinations. These entrenched beliefs are difficult to change in the short term, making identifying individuals who are more open to vaccination based on their attitudes essential. Therefore, there is an urgent need to develop a new computational model that efficiently addresses existing limitations and targets populations likely to accept vaccines.

This study aims to prioritize the subgroups that have the potential to obtain a COVID-19 vaccine in the future. We performed a cross-sectional study to investigate the vaccine hesitancy of Alabaman adults and developed the Embedding-based Spatial Information Gain (EMSIG) method to create a latent space that captures similarities based on perceptions of COVID-19 and trust levels, allowing us to identify subgroups. The central idea is that individuals with low or no COVID-19 vaccine doses who share similar vaccine perceptions and trust levels with those with high COVID-19 vaccine doses in the same subgroup are more likely to be open to change and get vaccinated in the future. Alabama, as a representative politically conservative state in the Deep South, serves as a valuable case study for examining vaccine hesitancy. By analyzing Alabama’s participants’ concerns through the machine learning model, EMSIG can explore barriers and inform targeted intervention strategies to potentially increase the COVID-19 vaccination rate in the Deep South. EMSIG advances three key aspects to fulfill our three objectives: (1) uncovering factors that naturally emerge from the data in this study, (2) identifying subgroups through the screening of regions of interest (ROI), and (3) revealing subgroup-specific concerns and characterizing the demographic features to target the identified subgroups effectively. First, EMSIG enhances factor extraction using a data-driven, correlation-based approach to evaluate global and local cluster quality, and it employs advanced large language models (LLMs) for factor annotation. Second, EMSIG uses embedding-based techniques to generate latent projections of participant perceptions and introduces spatial information gain (SIG) signals to quantify the predictive power of each factor and aggregated factors. Third, EMSIG conducts a disparity analysis to enrich demographic characteristics and establish relationships between these characteristics and perception factors. EMSIG represents an innovative approach for exploring subgroups within COVID-19 survey data, with implications for developing tailored and informative interventions.

## 2. Materials and Methods

### 2.1. Study Population and Data Collection

This cross-sectional study was conducted between 1 February and 11 March 2024 using an online structured questionnaire administered by Qualtrics, a survey sampling company, to Alabama residents aged 18 years and older. The survey targeted a diverse demographic representative of the state, employing sampling quotas based on race, ethnicity, COVID-19 vaccination status, and residency. Qualtrics sent out 3951 invitations and had 3101 (78%) survey entrants. After excluding 2081 responses due to various reasons, including quality terminations, in-survey terminations, over-quota terminations, and data quality checks, 1020 valid and complete responses remained. The questionnaire captured demographics, health status, and attitudes towards public health authorities and integrated the four validated measures to assess hesitancy towards COVID-19 vaccinations (CoVaH) [17] and trust across three dimensions: (1) medical doctors (T-DiG) [20], (2) pharmacists (Trust-Ph) [18], and (3) public health authorities (TiPHA) [19]. All scales demonstrated acceptable internal reliability as measured by Cronbach’s alpha, with a range of α = 0.753–0.958 across all the original scales and associated subscales.

### 2.2. The Data-Driven Correlation-Based Factor Extraction and Factor Annotation Based on the Large Language Model

We performed a five-step procedure for data-driven factor extraction based on all 85 items across the four primary scales (CoVaH, T-DiG, Trust-Ph, and TiPHA). Responses on the CoVah, T-DiG, and Trust-Ph ranged from 1 “Strongly disagree” to 5 ”Strongly agree”, with responses on the TiPHA ranged from 1 “Strongly agree” to 4 “Strongly disagree”.

First, we reversely scored items that were phrased in the opposite direction to the other items within each section. Second, we generated the questionnaire’s pairwise Pearson Correlation Coefficient (PCC) matrix using the formula pcc=∑ixi−x¯yi−y¯∑ixi−x¯2∑iyi−y2. We then performed agglomerative clustering to iteratively merge the most similar pairs of items until all items formed a single cluster in a tree structure, using Ward’s method to minimize the variance within clusters. Third, to determine the optimal number of clusters, we iteratively tested cluster sizes (k) ranging from 2 to 8 in the hierarchical cluster analysis with Ward’s linkage method and assessed the cluster quality based on both the Silhouette Coefficient (SC) [28] and the PCC within each cluster. The SC measures the relative cohesion of clusters based on intra-cluster and inter-cluster distances. For example, in a cluster (cI) with the center μcI, each item i∈cI’s in-cluster distance and the smallest mean distance to all points in any other cluster were calculated by a′i=di,μcI, and b′i=mincJ≠CI⁡di,μcJ, respectively. Then, the SC of the item *i* could be calculated by s′i=b′i−a′imax⁡{a′i,b′i}, and the cluster-wise SC could be calculated by SC′=maxk⁡1N∑is′i. The SC varied from −1 to 1, with higher values indicating better quality clusters. The in-cluster PCC mean-variance analysis, on the other hand, ensures the local cluster quality by measuring cluster cohesion and dispersion. To optimize the number of clusters within an acceptable range of the SC, we set the SC threshold above 0.25, which is the cutoff for “weak”. Meanwhile, we set a minimum threshold for the in-cluster PCC to ensure cluster quality. Thus, we derived the correlation-based factors from each scale by finalizing the cluster size. Fourth, we used LLMs to annotate the factors based on the survey questions. Particularly, we adopted ChatGTP-4 (https://chat.openai.com/, accessed on 11 July 2024), Gemini (https://gemini.google.com/, accessed on 11 July 2024), and meta-AI (https://www.meta.ai/, accessed on 11 July 2024) for the perception inference. The questions in each cluster were concatenated as input, and we generated prompt messages using the following template with the prompt strategy: “Identify one succinct aspect from the perceptions regarding vaccination in the related questions. \n [the survey questions]”. The LLMs returned key sentences in bold font. We manually reviewed the keywords generated by the three LLMs, selected the most representative word for factor annotation, and performed queries with the prompt five times for the repeated prompt strategy, selecting the most frequent response among the queries as the final result. Fifth, we calculated the factor scores by aggregating the item scores within each factor’s cluster.

### 2.3. The Participant’s Latent Projection Based on the 85-Item Questionnaire with a Five-Point Likert Scale

To explore the heterogeneity of factor scores using latent models, we utilized these factors and their scores as embeddings to project the participants into latent spaces using Uniform Manifold Approximation and Projection (UMAP) [29]. Latent non-linear projections often unravel complex combinations of scores in ways that are interpretable by humans. For example, a distance-based projection can capture the similarity between participants based on multi-factor scores. Specifically, we used all the factors extracted from the 85 questions to determine each participant’s latent projection. To visualize the heterogeneity in factor scales, we created a scatter plot of latent projections, applying a color scale to represent different factors and COVID-19 vaccine doses.

### 2.4. The Regions of Interest (ROI) Discovery to Form the Participant’s Subgroup in Latent Projection

To explore participant subgroups, we employed UMAP to generate a survey-based latent space and applied information gain to identify factors predicting COVID-19 vaccine doses. Particularly, we developed embedding-based spatial information gain (EMSIG) to explore the ROI where these subgroups were located. We were inspired by the spatial decision tree for geographical classification [30] to develop the spatial information gain (SIG) method to quantify the importance of a specific feature or features in refining the separation of COVID-19 vaccine doses in the latent space. Since the embedding space reflects survey similarity, an SIG for a particular factor suggests that it can effectively predict the COVID-19 vaccine doses within that ROI. Consequently, that factor can be used to inform and encourage participants in the identified subgroups to get the COVID-19 vaccination.

Firstly, we took the latent-projected coordinates in 2D space, denoted as $x_{i}$ and $y_{i}$, where i is the index of the participants. We pulled out the minimum and maximum values from $x_{i}$ and $y_{i}$, respectively, and created a grid matrix with a 0.5 step in the latent embeddings. Secondly, we performed subgroup screening based on the latent embedding. We took every grid position and calculated the grid-level’s SIG using the Kullback Leibler divergence formula $IG_{X,A}\left(X,a\right)=D_{KL}(P_{x}(x|a)||P_{X}(X|I))$, where X is a random variable, A and a are the same random variable before and after obtained for X, $P_{X}(X|I)$ is the prior distribution for X, and $P_{x}(x|a)$ is the posterior distribution for x given a. The expected value of the information gain is the mutual information *I*(*X*;*A*) of X and A. After calculating the SIG for all the pixels, we created an interpolated SIG signal surface based on the formula $h_{p|v}=h_{v}\frac{1}{\sqrt{2\pi }*\sigma }{exp}^{-\left(\frac{{\left\|d-v\right\|}_{2}}{2\sigma^{2}}\right)}$, where $h_{p|v}$ is the smoothed signal height at the pixel *p*, σ is a scale factor controlling the degree of signal dispersion (default is 1), and ${\left\|d-v\right\|}_{2}$ is the Euclidian distance from the position *v* to the current pixel *p*. Since the SIG signal cannot be negative, the record set $\left\{v_{p}\right\}$ in the ROI can be retrieved by raising the positive cutoff *c* in the interpolated SIG signal surface in the formula $\left\{v_{p}\}∊\{h_{p|v}\geq c\right\}$. Thirdly, we set the cutoff *c* to be 34% of the maximum value of the SIG signals to filter the subgroups with clear boundaries. The contour lines were added based on the four levels of the SIG signals: 0%, 17%, 34%, and 51%.

### 2.5. The Correlation Analysis Between the Factor and COVID-19 Doses in Each ROI Subgroup

To reveal the correlation between each factor and COVID-19 doses in the ROI subgroups, we performed Pearson Correlation analysis and calculated the PCC. We set the absolute PCC cutoff at 0.2 to identify at least weak correlations between COVID-19 vaccine doses and the factor.

### 2.6. The Statistical Analysis of the Patient’s Characteristics in ROI Subgroups

To investigate disparities among patients with distinct characteristics across different ROI subgroups, we conducted a chi-squared contingency test using the SciPy library in Python. Specifically, we used the frequency of each category in a selected variable among all participants as the null hypothesis ($H_{0}$). We then considered the observed frequency of each category in the selected variable within the ROI subgroup as the observation ($O_{i}$, where *i* is the index of categories). The expected frequency of each category was based on the whole population ($E_{i}$). The chi-square statistic was calculated using the formula $\chi^{2}=\sum_{i=1}^{k}\frac{{\left(O_{i}-E_{i}\right)}^{2}}{E_{i}}$, where k is the total number of categories, and the degree of freedom is k−1. We applied a *p*-value cutoff of 0.05 to filter statistically significant results. When a category had too few observations, we pooled the categories with a sample size larger than 5. Subsequently, we extracted these significant characteristics and visualized them using pie charts to compare the proportions of these characteristics across the ROI subgroups.

### 2.7. The Regression-Based Machine Learning Model for COVID-19 Vaccine Dose Prediction

We performed and compared supervised ML models in the six major categories, including (1) Linear/generalized linear methods (Linear Regression [31], Least Angle Regression [32], Huber Regressor, Elastic Net [33], Orthogonal Matching Pursuit [34], and Passive Aggressive Regressor [35]); (2) Tree-based methods (Decision Tree Regressor [36]); (3) Instance-based methods (K Neighbors Regressor [37]); (4) Bayesian methods (Bayesian Ridge [38]); (5) Gradient boosting methods (Gradient Boosting Regressor [39], Light Gradient Boosting Machine [40], and Extreme Gradient Boosting [41]); and (6) Ensemble methods (Random Forest Regressor [42], Extra Trees Regressor [43], and AdaBoost Regressor [44]). To evaluate the performance of the candidate models, we applied the coefficient of determination (R^2^), indicating the goodness of fit [45]. We also measured five additional metrics, including Mean Absolute Error (MAE), Mean Squared Error (MSE), Root Mean Squared Error (RMSE), Root Mean Squared Logarithmic Error (RMSLE), and Mean Absolute Percentage Error (MAPE). The 1020 participants were divided into 70% for training and 30% for testing. The 70% training set was further subdivided, with 63% for training and 7% for validation within a 10-fold cross-validation framework. The model’s performance was evaluated based on the testing set. In addition, we utilized two measurements, feature importance [46] and Shapley values [47], to extract the predominant predictors. Feature importance measures each feature’s relative contribution in reducing the training dataset’s variance. Shapley values measure the contribution of individual predictors in a regression model based on cooperative game theory.

## 3. Results

### 3.1. Sample Descriptives

The sample consisted of 1020 participants (Table 1), with 29.9% identifying as male and 70.1% as female. The majority of participants were White (69.6%), followed by Black (21.8%), multi-racial (4.2%), and smaller proportions were Asian (1.1%) and other races (3.9%). Most participants (93.0%) identified as not Hispanic or Latino. The age distribution ranged from 18 to over 65, with the largest group being 45 to 54 years old (19.9%). Nearly half (49.3%) had a high school diploma or GED as their highest degree obtained. The sample had a higher proportion of unmarried participants (58.5%), and 44.0% were employed. In terms of household income, 38.0% of participants reported earning $0–$30,000 annually. Most were not caregivers (85.3%) and were insured (92.7%). Regarding health status, 36.8% reported being in good health. The sample had mixed political affiliations, with 41.7% identifying as Republican and 20.5% as Democrat. Self-reported vaccination statuses showed that 58.2% had ever received a COVID-19 vaccine, while 39.5% had received an influenza vaccine in the 2023–2024 season. Further delineating COVID-19 vaccination uptake, 24.3% reported receiving a total of two doses, 12.1% reported three doses, and smaller proportions reported receiving four (6.4%) or five or more doses (5.6%)—7.8% of participants reported ever receiving only one dose.

### 3.2. The 16 Factors Extracted from the 85-Item Questionnaire with a Five-Point Likert Scale

The 16 factors extracted from the total of 85 items across the CoVaH, T-DiG, Trust-Ph, and TiPHA measures balanced global and local cluster quality, with unbiased annotation facilitated by LLMs (Figure 1). The multidimensional scale of the survey was adopted to assess the participants’ concerns about taking the COVID-19 vaccination. The data-driven factor extraction provided new insights into these concerns through the four scales, highlighting their heterogeneity. We found four factors in the CoVaH questionnaire, including (C-c1) detriment, (C-c2) fear, (C-c3) skepticism, and (C-c4) side effects; five factors in the T-DiG scale, including (D-c1) trust, (D-c2) fairness, (D-c3) accountability, (D-c4) motivation, and (D-c5) impartiality; three factors in Trust-Ph scale, including (P-c1) communication, (P-c2) expertise, and (P-c3) trust; and four factors in the TiPHA scale, including (G-c1) trust, (G-c2) intelligence, (G-c3) fairness, and (G-c4) responsibility. The factor annotations were generated by the prompt’s summarization in LLMs (Figure 1). The SC was set to be no less than 0.25 in each scale. We determined the optimal cluster number by ensuring that local PCC scores did not decline as the SC increased and that the PCC variance remained minimized. The optimal cluster number for CoVaH is 4, based on a minimum PCC of at least 0.7 and the variance of the PCC reaching its minimum value. Similarly, the optimal cluster number for T-DiG is 5, with a minimum PCC of at least 0.6. The optimal cluster number for TRUST-Ph is 3, with a minimum PCC of at least 0.5. Lastly, the optimal cluster number for Trust in Public Health Authorities is 4, with a minimum PCC of at least 0.6 (Appendix A). Based on the distribution of the scores for each factor, the majority of the factors followed a normal distribution or bimodal distribution. However, the COVID-19 detriment factor showed high divergence, with high frequencies in low and high scores. The COVID-19 fear factor was not a primary concern compared to COVID-19 detriment, skepticism, and side effects, as a high proportion of participants disagreed with it. Factors such as doctor trust, fairness, accountability, pharmacist trust, and government trust were generally deemed plausible, as participants were likely to agree with them. By employing a data-driven, correlation-based approach, we identified 16 factors that achieved an optimal balance between global and local cluster quality, specific to our multidimensional scale survey.

### 3.3. The Random Forest Regressor Performed the Best for the COVID-19 Vaccine Dose Regression Using the 16 Factors

The random forest regressor was the best-performing machine learning model to predict the COVID-19 vaccine doses based on the factors (Table 2). We conducted regression-based supervised learning and used six metrics to evaluate the performance of the regression models. The random forest regressor outperformed the others, demonstrating the smallest errors according to the mean absolute error (MAE), mean squared error (MSE), and root mean squared error (RMSE). Additionally, the R^2^ value for the testing dataset using the random forest regressor was 0.45, indicating that this model could explain 45% of the variance in the data. Based on the random forest regressor model, the three predominant factors were COVID-19 perceived detriment, fear, and side effects, as determined by the prioritization by both Shapley values and feature importance (Appendix A). Particularly, the top five factors prioritized by Shapley values were COVID-19 perceived detriment, perceived fear, side effects, skepticism, and trust in doctors, as they accounted for the largest marginal contribution to the variance of the COVID-19 vaccine doses. In the sensitivity analysis to evaluate the robustness of our results by resampling the data with 10 different seed values, the COVID-19 perceived detriment, fear, and side effects remained in the top 5 list. Therefore, the three key concerns, COVID-19 perceived detriment, fear, and side effects, prioritized by the machine learning model, play more essential roles than other factors across the entire population.

### 3.4. The Four ROI Subgroups Extracted Based on the SIG Signal in Latent Projection and the Demographic’s Disparity Analysis

The latent projection helped explain the variations in COVID-19 vaccine doses by revealing highly variable dose patterns correlated with differing combinations of the 16 factors (Figure 2). This projection was generated using scores from the 16 factors. Overall, participants with higher scores in COVID-19 vaccine hesitancy questions indicated a higher degree of disagreement with these factors, corresponding to a higher vaccination dose. Conversely, participants with higher scores on trust questions indicated high disagreement with the trust factors, corresponding to a lower vaccine dose. Participants located at the center or bottom-right regions tended to be relatively conservative, as indicated by their low mean and variance in vaccine doses, implying they were resistant to the COVID-19 vaccination and were reluctant to change their minds easily. However, certain regions at the top, top left, and bottom of the latent space exhibited a large variance in COVID-19 vaccine doses despite showing similar survey results in their latent embedding. These subgroups were particularly interesting, as the participants with low/no COVID-19 vaccine doses might be more likely to get vaccinated if their specific concerns are addressed. Therefore, the multiple-scale factors of COVID-19 vaccine hesitancy and trust scales generated latent projections, revealing regions with high variability in COVID-19 vaccine doses, each characterized by distinct factor patterns.

The ROI subgroups displayed significant heterogeneity in demographic characteristics and considerable variance in COVID-19 vaccine doses in each subgroup (Figure 3). We identified four ROI subgroups: ROI-1 with a subgroup of 112 participants, ROI-2 with a subgroup of 42 participants, ROI-3 with a subgroup of 145 participants, and ROI-4 with a subgroup of 21 participants, based on the SIG signal extraction. Each factor’s SIG signal exhibited heterogeneity and consensus (Appendix A). Four ROI subgroups emerged after aggregating the SIG signals from the four scales and applying the signal contour line as 34% of the maximum signal (Figure 3a). After performing the chi-square contingency test, 12 demographic characteristics—gender, race, ethnicity, age, education, marital status, employment, income, insurance, political affiliation, flu shot history, and health status—showed significance in at least one of the ROI subgroups (Figure 3b). The average vaccine dose among all 1020 participants was 1.5. The ROI-1 subgroup had an average of 2.3 doses, the ROI-2 subgroup had 1.6 doses, the ROI-3 subgroup had 2.6 doses, and the ROI-4 subgroup had 1.7 doses. According to the proportional changes, the ROI-1 subgroup, characterized by a higher flu shot vaccination rate (61.6%), had a higher proportion of individuals reporting excellent health status (20.5%) compared to all participants (11.8%) (Figure 3c). The ROI-2 subgroup had a higher proportion of white (95.2%), older (45.2% participants were 65 and above), and Republican (81%) participants. The ROI-3 subgroup had a higher proportion of Democrats (38.6%), with a high flu shot rate of 59.3%. Additionally, the ROI-3 subgroup had a higher proportion of participants with a bachelor’s degree or higher (29%) compared to all participants (20%). The ROI-4 subgroup had a higher proportion of minorities and a flu shot rate at the baseline level of 33.3%. The common factors shared by at least three of the four ROI subgroups were perceived detriment, fear, skepticism, side effects related to COVID-19, and communication with pharmacists. The type of insurance and marital status significantly differed in the ROI-2 subgroup due to the association of these characteristics with age, with a higher proportion of married individuals and Medicare plan coverage for people 65 and older. The income in the ROI-4 subgroup was significantly different, with a higher proportion of individuals choosing “prefer not to say”. Therefore, EMSIG identified four ROI subgroups with distinct concerns and revealed their specific demographic characteristics for potential tailored interventions.

## 4. Discussion

We conducted a data-driven, correlation-based analysis of COVID-19 vaccination attitudes using an 85-item questionnaire completed by 1020 participants in Alabama and identified 16 key factors which differed from what has been previously reported. Using our data, we identified: “detriment”, “fear”, “skepticism”, and “side effects”. These factors differ slightly from the three identified in the published CoVaH [17] study (“skepticism”, “risk”, and “fear”). For five questions—regarding the perceived goodness, importance, community benefit, effectiveness, and provider recommendation of the COVID-19 vaccine—we summarized the positive responses as “benefit” and used reverse coding to classify negative responses as “detriment” rather than “skepticism” as identified by CoVaH. The CoVaH’s “risk” was divided into two factors: “skepticism” and “side effects”. The ‘skepticism’ factor consisted of two questions: one regarding the perception that the government created COVID-19 and the other expressing caution about government recommendations. Next, the published T-DiG reported seven factors [20], including “communication competency”, “fidelity”, “system trust”, “confidentiality”, “fairness”, “stigma-based discrimination”, and “global trust”. For our study, five factors were extracted. That is, “trust” was a result of the merging of “global trust” and “communication competency”. The factor “accountability” was formed by combining “system trust” and “confidentiality”. We derived the factor “motivation” by reverse coding “fidelity”, and “impartiality” was formed by reverse coding “stigma-based discrimination”. Compared to the three factors previously reported for the TRUST-Ph [18] scale which consists of: “benevolence”, “technical competence”, and “communication”, we extracted four factors. The “expertise” factor consists of three questions: “Pharmacists should be the persons who make decisions about your medications”, “Pharmacists can help you with your illness”, and “Pharmacists can solve your medication problems”. These were identified as a subset of the original factor of “technical competence”. The remaining questions from “technical competence” were combined with “benevolence” to form the factor “communication”. Lastly, the TiPHA [19] reported two factors while our study extracted four factors. Specifically, the factor “beneficence” was replaced by the data-driven correlation-based factors “trust”, “intelligence”, and “fairness”. The questions regarding “unhelpful recommendations” and “wasting money” formed the factor “responsibility”. For the factor annotations, LLMs offered significant advantages for inferring the key concept as a factor annotation due to their contextual understanding [48]. The representative words extracted from the integrated prompts results of the three state-of-the-art LLMs addressed the subjectivity inherent in the factor labels.

To identify subgroups of Alabama residents with similar perceptions, enriched by demographic characteristics but with varying COVID-19 vaccine doses, we developed the EMSIG to explore the ROI subgroups. The ROI subgroups were critical for identifying participants who have the potential to obtain additional COVID-19 vaccine doses, given that their survey answers were similar to those of high-dose individuals but with minor concerns that could be shown by EMSIG. The four ROI subgroups exhibited distinct demographic characteristics. The ROI-1 subgroup, characterized by relatively good health status, showed similarities to the ROI-3 subgroup characterized by educated Democrats, which had a high flu-shot rate. In comparison, the ROI-2 subgroup was characterized by being white, older, and Republican. Lastly, the ROI-4 subgroup was characterized by minority status and a relatively lower flu shot rate. This finding supports the idea that political preference differences influence vaccination attitudes via political networks [6,7,8,9,10,11,12].

This study addressed the barriers to the uptake of the COVID-19 vaccination in various groups and identified factors critical to the success of multi-faceted interventions targeted to overcome vaccine hesitancy [25]. To encourage participants, particularly those with low COVID-19 vaccine doses in the ROI subgroups with high COVID-19 variance, several common concerns need to be addressed, including the perceived detriments, fears, and side effects of COVID-19. Notably, the vaccination efficacy and effectiveness factors included in the COVID-19 perceived detriment factor have been reported in other studies [15,21,22]. Meanwhile, the trust factors, including trust in corporations, trust in health authorities, and healthcare provider advice reported previously [15,21,22,23] have been found in each ROI-specific subgroup as specific concerns. The ROI-1 and ROI-3 subgroups tend to put “pharmacists communication” and “government responsibility” as the top priority, specifically. In contrast, the ROI-2 subgroup showed more concerns with the “doctor trust” and “government intelligence” specifically. Therefore, to persuade the ROI-1 and ROI-3 subgroups to get the COVID-19 vaccination, it is crucial to highlight the helpful recommendations of public health agencies as well as how they address drug problems and promote efficient spending practices. Additionally, endorsements and recommendations from pharmacists should be emphasized. To persuade the ROI-2 subgroup to get the COVID-19 vaccination, it is crucial to solve their concerns about doctor trust and the intelligence of public health authorities, especially regarding the repeated strategies to help the public despite acknowledging their limited effectiveness. The concerns regarding pharmacist communication expressed by the ROI-1 and ROI-3 subgroups highlight the critical role of pharmacists as advocates for COVID-19 vaccination, as well as educators and vaccine administrators. By adhering to the regulations of their respective jurisdictions, pharmacists can significantly contribute to vaccination efforts [49]. Restoring trust in doctors for the ROI-2 subgroup, characterized by being white, older, and Republican participants, is crucial for improving vaccination compliance and other health behaviors, especially given the survey’s findings of a significant decline of trust in physicians and hospitals during the COVID-19 pandemic [50]. Health and political leaders should work to enhance public trust in the government [51], particularly addressing the concerns of the ROI-2 subgroup regarding intelligence, as well as the responsibilities highlighted by the ROI-1 and ROI-3 subgroups, to potentially increase vaccination rates. Providers can offer trusted and reliable resources to families, including local and national public health and government websites as well as fair and unbiased social media sites that provide transparent, accurate, and updated information [52] to address the fundamental concerns regarding perceived detriment, fear, skepticism, and side effects. For instance, the Centers for Disease Control and Prevention (CDC) provides timely updates for research on adverse events following COVID-19 vaccination [53]. Additionally, it is not surprising that the ROI-1 and ROI-3 subgroups exhibited both higher COVID-19 vaccine doses and flu-shot rates compared to the ROI-2 and ROI-4 subgroups, as another study has also demonstrated a strong association between flu-shot updates and COVID-19 vaccine uptake [54].

The current research has several limitations that need to be addressed in future studies. First, we did not consider the brands of the COVID-19 vaccination (e.g., Moderna, Pfizer, Johnson & Johnson) and their impact on vaccine hesitancy. Different vaccines are perceived differently by the public and vary in terms of side effects and efficacy, all of which could affect participants’ responses and their willingness to get vaccinated. Second, for the number of COVID-19 vaccine doses, since we did not request the dates of each COVID-19 vaccination, we could not identify when each vaccine dose was received. Additionally, we did not account for individuals who did not receive the vaccination due to prior COVID-19 infection. Addressing these gaps could enhance our model for targeting individuals susceptible to COVID-19, considering that the effectiveness of COVID-19 vaccines tends to decline after six months [55,56]. Third, to validate our discovery of the factors and their influence on COVID-19 vaccine doses, our future work will focus on developing multi-faceted interventions to increase COVID-19 vaccine uptake. Fourth, this study is limited to the state of Alabama. Expanding the research to cover the Deep South or conducting a national study could reveal subgroup perceptions and heterogeneity of demographic characteristics. At the national level, political influence on COVID-19 vaccine dose variance can be more effectively analyzed by comparing party affiliations across different states. Since the Deep South lags in coronavirus vaccination rates and experiences significant racial and ethnic residential segregation [5], examining ecological and spatiotemporal differences (e.g., urban vs. rural) could provide valuable insights for tailored interventions, especially in comparison to other regions. Fifth, since this was a cross-sectional study, causality cannot be established. Sixth, while our model identified subgroup-specific factors, it may not capture all nuances across diverse populations. For instance, the levels of health literacy [57,58,59] can vary significantly. The model did not efficiently evaluate the main concerns specified through open-ended questions expressed by the participants. Additionally, the current design has not incorporated a longitudinal study, and the modeling techniques could help validate the findings and explore changes in attitudes over time. Seventh, the EMSIG model could be enhanced by integrating multidimensional data to create comprehensive latent embeddings. This approach would allow for a more holistic representation of factors, capturing demographics, vaccine hesitancy, trust scales, and geographic information, ultimately improving the model’s accuracy and ability to identify subgroup-specific concerns. Lastly, the survey participants were subject to selection bias and recall bias, which may have influenced the EMSIG results with distorted data. The EMSIG method could be applied to other vaccines, such as the flu shot, to explore subgroup concerns and develop targeted interventions.

## 5. Conclusions

The Deep South, U.S., continues to have a low COVID-19 vaccination rate. The COVID-19 vaccination hesitancy and trust scales are essential to assess the concerns of participants with low or no COVID-19 vaccine uptake. To further target the subgroups with the potential to change their attitudes and eventually get COVID-19 vaccination based on their perception of COVID-19 and trust levels, we developed a machine learning technique named EMSIG. EMSIG identified four ROI subgroups where participants exhibited diverse concerns. Common concerns included the perceived harm of COVID-19, fear, and potential side effects, which need to be addressed as fundamental issues. Additionally, a subgroup primarily composed of educated Democrats with a high flu-shot rate expressed concerns about pharmacist communication, government fairness, and responsibility. Another subgroup, characterized by older, white Republicans with a relatively low flu-shot rate, expressed concerns about doctor trust and the intelligence of public health authorities, particularly regarding the effectiveness of their strategies for supporting public health.

## Figures and Tables

**Figure 1 vaccines-12-01253-f001:**
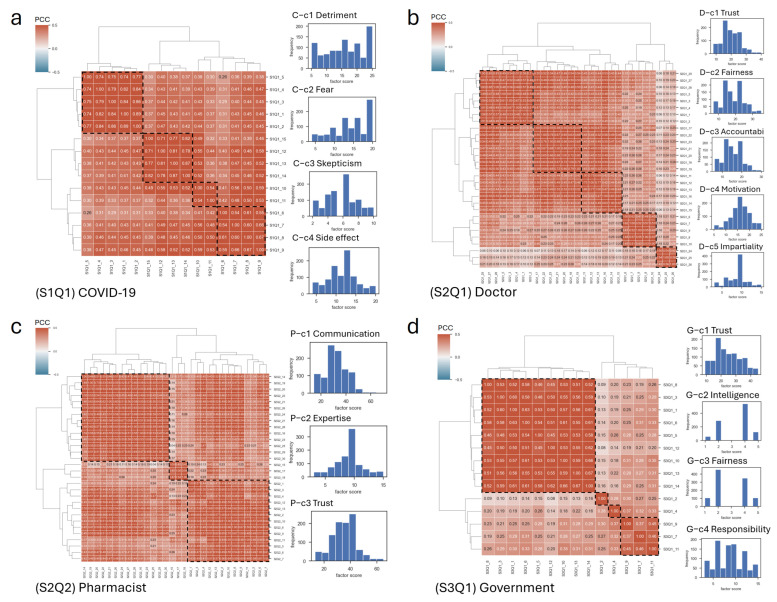
The heatmap of the Pearson Correlation Coefficient (PCC) matrix and the agglomerative clustering. The correlation-based factors were extracted from (**a**) COVID-19 Vaccine Hesitancy, (**b**) Trust in Doctors, (**c**) Trust in community Pharmacists, and (**d**) Trust in Public Health Authorities. The data-driven correlation-based factor annotation is based on the large language models and the distribution of the factor scores. The rows and columns in the heatmap represent the questions in the 85-item questionnaire. The dashed frame represents the questions grouped together to form the factor in each scale. The hierarchical tree on the side of the heatmap represents the agglomerative clustering with Ward’s method to form clusters. The histograms on the right side of each heatmap illustrate the distribution of factor scores for each specific factor. The factor annotations were based on the prompt results of the LLMs, as described in Methods in Section 2.2. The C-c1 represents the COVID-19 cluster 1.

**Figure 2 vaccines-12-01253-f002:**
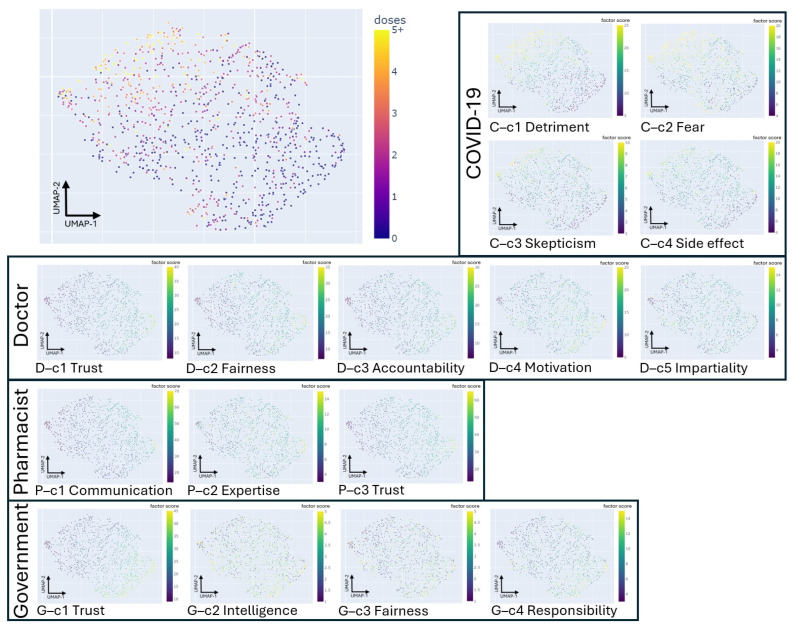
The participant’s latent projection in 2D space. The Uniform Manifold Approximation and Projection (UMAP) plot on the upper left corner represents the latent projection overlay of the COVID-19 vaccine doses with the plasma-style color scale. The UMAP plots in the solid black boxes represent the overlap of factor scores with the viridis-style color scale. Higher scores on trust questions indicated high disagreement with the trust factors.

**Figure 3 vaccines-12-01253-f003:**
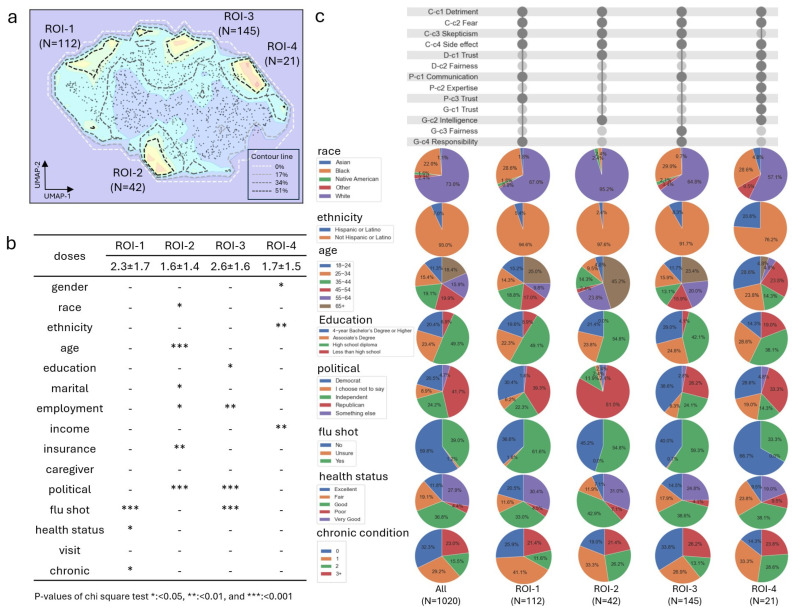
The ROI subgroup extraction and the disparity analysis. (**a**) The ROI subgroup extraction based on the aggregated embedding-based spatial information gain (EMSIG) signals using Uniform Manifold Approximation and Projection (UMAP). The black dot represents a participant’s latent projection in the embedding space. The color gradient from blue to red indicates the range of spatial information gain, with blue representing low values and red representing high values. The dashed lines in the latent space represent the contour lines of the SIG signal cutoffs, represented by the percentages of the maximum signal. The contour line cutoff was set to be 34% of the maximum value of the SIG signals to filter the four ROI subgroups. (**b**) The chi contingency test for factor representation in each ROI subgroup compares all the participants. The COVID-19 vaccine doses are denoted as the mean ± one standard deviation. The chi-square test’s *p*-value of 0.05 was used as the cutoff to filter the significant characteristics of the four ROI subgroups. The symbols “-”, “*”, “**” and “***” represent significance levels based on *p*-value cutoffs. (**c**) The representative factor and the demographic characteristics in each ROI subgroup. The dark black nodes indicate factors that have at least a weak correlation with COVID-19 vaccine doses. The pie charts show the composition of demographic categories for each demographic characteristic across all participants and the four ROI subgroups.

**Table 1 vaccines-12-01253-t001:** Participant Demographics and Characteristics (*n* = 1020).

Variable	N (%)
Sex	
Male	305 (29.9)
Female	715 (70.1)
Race	
White	710 (69.6)
Black	216 (21.8)
Asian	11 (1.1)
Other	40 (3.9)
Multi-racial	43 (4.2)
Ethnicity	
Hispanic or Latino	71 (7.0)
Not Hispanic or Latino	949 (93.0)
Age	
18–24	115 (11.3)
25–34	157 (15.4)
35–44	195 (19.1)
45–54	203 (19.9)
55–64	162 (15.9)
65+	188 (18.4)
Highest Degree Obtained	
Less than high school	70 (6.9)
High school diploma or GED	503 (49.3)
Associate’s degree or Vocational Certificate	239 (23.4)
4-year Bachelor’s Degree or Higher	208 (20.4)
Marital Status	
Married	423 (41.5)
Not Married	597 (58.5)
Employment Status	
Employed	449 (44.0)
Not Employed	199 (19.5)
Retired	211 (20.7)
Disabled, not able to work	161 (15.8)
Household Income Level	
$0–$30,000	388 (38.0)
$30,001–$60,000	313 (30.7)
$60,001–$90,000	142 (13.9)
$90,001–$120,000	60 (5.9)
$120,000+	62 (6.1)
I choose not to say	55 (5.4)
Caregiver Status	
Caregiver	150 (14.7)
Not Caregiver	870 (85.3)
Political Affiliation	
Republican	425 (41.7)
Democrat	209 (20.5)
Independent	247 (24.2)
Something Else	48 (4.7)
I choose not to say	72 (7.3)
Insurance Status	
Insured	913 (92.7)
Not Insured	72 (7.3)
Health Status ^	
Excellent	120 (11.8)
Very Good	285 (27.9)
Good	375 (36.8)
Fair	195 (19.1)
Poor	45 (4.4)
Total Number of Chronic Conditions Endorsed	
0	329 (32.3)
1	298 (29.2)
2	158 (15.5)
3	125 (12.3)
4 or more	108 (10.6)
Missing	2 (0.2)
Ever Received COVID-19 Vaccination	
Yes, Received	573 (56.2)
No, Did Not Ever Receive	447 (43.8)
Total Number of COVID-19 Doses Received	
0	447 (43.8)
1	80 (7.8)
2	248 (24.3)
3	123 (12.1)
4	65 (6.4)
5 or more	57 (5.6)
Received Influenza Vaccination in 2023–2024	
Yes, Received	398 (39.5)
No, Did Not Receive	610 (60.5)

^: The self-rated overall health status refers to Section 4, Question 2 under the General Health subsection in Appendix A.

**Table 2 vaccines-12-01253-t002:** The regression-based supervised learning in the COVID-19 vaccination prediction on the testing set. The 19 regression models were ranked based on the R^2^. The highest performance value for each evaluation metric is highlighted in bold text. R^2^: Coefficient of Determination, MAE: Mean Absolute Error, MSE: Mean Squared Error, RMSE: Root Mean Squared Error, RMSLE: Root Mean Squared Logarithmic Error, and MAPE: Mean Absolute Percentage Error.

Model	R^2^	MAE	MSE	RMSE	RMSLE	MAPE
Random Forest Regressor	**0.45**	**0.90**	**1.35**	**1.16**	0.52	0.43
Bayesian Ridge	0.43	0.94	1.39	1.18	0.53	0.39
Linear Regression	0.42	0.94	1.40	1.18	0.53	0.40
Ridge Regression	0.42	0.94	1.40	1.18	0.53	0.40
Extra Trees Regressor	0.42	0.92	1.41	1.18	0.52	0.44
Elastic Net	0.42	0.94	1.40	1.18	0.53	0.39
Least Angle Regression	0.42	0.94	1.41	1.19	0.53	0.40
Huber Regressor	0.41	0.94	1.43	1.19	0.53	0.41
Lasso Least Angle Regression	0.41	0.95	1.44	1.20	0.54	0.38
Lasso Regression	0.41	0.95	1.44	1.20	0.54	0.38
Gradient Boosting Regressor	0.41	0.92	1.44	1.20	**0.51**	0.45
K Neighbors Regressor	0.40	0.92	1.47	1.21	0.54	0.46
Orthogonal Matching Pursuit	0.40	0.95	1.45	1.21	0.54	0.40
Light Gradient Boosting Machine	0.39	0.93	1.49	1.22	0.53	0.47
AdaBoost Regressor	0.36	1.05	1.55	1.24	0.59	**0.37**
Extreme Gradient Boosting	0.29	1.00	1.74	1.31	0.56	0.50
Dummy Regressor	−0.02	1.38	2.51	1.58	0.70	0.44

## Data Availability

The data presented in this study are available on request from the corresponding author. The data are not publicly available due to potential breaches of confidentiality, particularly in areas with few participants, and the possibility that openly sharing the data would jeopardize our ability to conduct future research using the same dataset.

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
