# Peer review of "EMSIG: Uncovering Factors Influencing COVID-19 Vaccination Across Different Subgroups Characterized by Embedding-Based Spatial Information Gain"

_vaccines, 2024, doi:10.3390/vaccines12111253_

Round 1
Reviewer 1 Report
Comments and Suggestions for Authors
- Lines 19-20: There is more recent data on this. This is the only instance where this information on burden is provided, it should be included in the Introduction and backed up by a reference.
- Lines 48-50: How can a reference published in 2019 about attitudes towards childhood vaccinations be used to explain disparities in COVID-19 vaccination and booster rates? The body of evidence on COVID-19 vaccine acceptance is large and such studies should be used accordingly.
- Lines 52-54: Same as the above comment. It would be appropriate to state that a study done then and then about that and that showed that and that and that is the reason why something might be considered about the present situation with COVID-19 vaccination. Otherwise it's misleading the reader to think that the cited references are exactly and specifically on the mentioned topic. There are many studies on the effects of political leaning on COVID-19 vaccine hesitancy.
- Lines 56-58: Does the reference No. 7 even mention vaccines? Were vaccines even available at that point when that article was published? In the title it even states that it is about social distancing. The accuracy of citing references to back-up the claims throughout the entire manuscript needs to be double checked.
- Lines 59-79: Does this text belong in the Methods?
- Introduction: Do the last three paragraphs refer to the applied methodology or provide an overview of previous work in the field with the purpose of highlighting the need for this study? As it currently reads, the Introduction section needs to be improved to better explain the rationale for this study.
- Line 117: The 1020 individuals participating out of how many contacted, i.e. what is the response rate to the survey?
- Methods: Description of study's power is missing.
- Line 117: How was COVID-19 status defined?
- Line 118: Variables need to be defined. For example, how was health status investigated and how are different categories of health status as shown in Table 1 defined?
- Line 301: Should this cite Table 2 instead of Table 1?
- Lines 302-303: The performance metrics should be stated in the methods section.
- Lines 313-314: Where is the red marking?
- Methods: How were missing data handled?
- Discussion: Discussion cites about 5 references, are there no more studies that provide findings that could be used to provide explanations for findings? A more comprehensive overview of the available literature is necessary to discuss the findings.
- Supplementary materials that are mentioned are not available anywhere for review.
- Conclusion is missing and needs to be added.
- How come an IRB exemption was provided for conducting a cross-sectional study? What are the criteria? Additionally, how come a waiver of documentation of consent was used? Is this also determined in the exempt decision?
Author Response
|
Response to Reviewer 1’s Comments
|
- Summary
We appreciate the reviewer's comments and suggestions. Please find the detailed responses below and the corresponding revisions/corrections highlighted/in track changes in the re-submitted files.
|
2. Questions for General Evaluation |
Reviewer’s Evaluation |
Response and Revisions |
|
Does the introduction provide sufficient background and include all relevant references? |
Must be improved |
|
|
Are all the cited references relevant to the research? |
Can be improved |
|
|
Is the research design appropriate? |
Must be improved |
|
|
Are the methods adequately described? |
Yes |
|
|
Are the results clearly presented? |
Must be improved |
|
|
Are the conclusions supported by the results? |
Must be improved |
|
|
3. Point-by-point response to Comments and Suggestions for Authors |
Comments 1: Lines 19-20: There is more recent data on this. This is the only instance where this information on burden is provided, it should be included in the Introduction and backed up by a reference.
Response 1: We’ve updated the COVID-19 death count to the most recently released data (up to 9/29/2024) from the source “Our World in Data”. We’ve also included the reference in the introduction section. (Lines 14 and 42)
Comments 2: Lines 48-50: How can a reference published in 2019 about attitudes towards childhood vaccinations be used to explain disparities in COVID-19 vaccination and booster rates? The body of evidence on COVID-19 vaccine acceptance is large and such studies should be used accordingly.
Response 2: The 2019 reference was initially included to illustrate the strong partisan motivations influencing individuals' reception and understanding of information. However, I understand your concern regarding its relevance to COVID-19 vaccination attitudes. Therefore, we have replaced it with more direct evidence from a newer reference specifically addressing COVID-19 vaccine acceptance. (Line 53)
Comments 3: Lines 52-54: Same as the above comment. It would be appropriate to state that a study done then and then about that and that showed that and that and that is the reason why something might be considered about the present situation with COVID-19 vaccination. Otherwise it's misleading the reader to think that the cited references are exactly and specifically on the mentioned topic. There are many studies on the effects of political leaning on COVID-19 vaccine hesitancy.
Response 3: We have reorganized the sentences, added additional references, and provided a concise conclusion to clarify the party affiliation and limited knowledge of COVID-19 on the vaccine hesitancy. (Lines 54-64)
Comments 3: Does the reference No. 7 even mention vaccines? Were vaccines even available at that point when that article was published? In the title it even states that it is about social distancing. The accuracy of citing references to back-up the claims throughout the entire manuscript needs to be double checked.
Response 3: The reference does not specifically mention vaccines; instead, it illustrates how political ideology influences behaviors and attitudes towards COVID-19 virus. We have replaced it with more direct evidence that highlights the influence of partisanship on COVID-19 vaccination. (Line 59)
Comments 4: Lines 59-79: Does this text belong in the Methods?
Response 4: We’ve revised the paragraph and retained it in the introduction section, as these scales were developed in previous work, and we believe it is more appropriate to provide a descriptive introduction for them here. (Lines 86-90)
Comments 5: Introduction: Do the last three paragraphs refer to the applied methodology or provide an overview of previous work in the field with the purpose of highlighting the need for this study? As it currently reads, the Introduction section needs to be improved to better explain the rationale for this study.
Response 5: We've revised the last three paragraphs of the introduction to better illustrate the purpose of the study and provide a detailed explanation of the rationale behind it. (Lines 113-129)
Comments 6: Line 117: The 1020 individuals participating out of how many contacted, i.e. what is the response rate to the survey?
Response 6: We had a contract with Qualtrics to collect a set quota of 1,000 responses. Qualtrics was responsible for recruiting participants and ensuring that the sample met our inclusion criteria and was representative of Alabama’s population in terms of race, ethnicity, and vaccination status. Qualtrics sent out 3,951 invitations and had 3,101 survey entrants. After excluding 2,081 responses due to various reasons including quality terminations, in-survey terminations, overquota terminations, and data quality checks, 1,020 valid and complete responses remained.
Comments 7: Methods: Description of study's power is missing.
Response 7: We did not do any power calculation because the primary purpose is exploratory in nature. Instead, we calculated the number of desired responses based on the number of AL adult population of 4 M, the confidence level of 95, and a margin of error of 3.
Comments 8: How was COVID-19 status defined?
Response 8: To enhance clarity, we replaced “COVID-19 status” with “COVID-19 vaccination status”, indicating whether participants received the COVID-19 vaccine. The quota was set to 50% vaccinated and 50% unvaccinated.
Comments 9: Line 118: Variables need to be defined. For example, how was health status investigated and how are different categories of health status as shown in Table 1 defined?
Response 9: To improve clarity, we’ve added a brief introduction to the health status and referenced the detailed information in Section 4, Question 2 under the General Health subsection in Supplementary File 1.
Comments 10: Line 301: Should this cite Table 2 instead of Table 1?
Response 10: Thanks. We’ve updated it. (Line 345)
Comments 11: Lines 302-303: The performance metrics should be stated in the methods section.
Response 11: We’ve added the metrics in the methods section. (Lines 268-274)
Comments 12: Lines 313-314: Where is the red marking?
Response 12: We replaced the red text with bold formatting to enhance clarity. (Line 345)
Comments 13: Methods: How were missing data handled?
Response 13: In the 85-item questionnaire, no missing data was found for any responses after the quality check. However, for the question regarding chronic conditions, participants with missing responses were retained in the analysis.
Comments 14: Discussion: Discussion cites about 5 references, are there no more studies that provide findings that could be used to provide explanations for findings? A more comprehensive overview of the available literature is necessary to discuss the findings.
Response 14: We’ve added more studies with citations to support the findings in the discussion section. (Lines 475, 490, 498-499, 518-519, 531, and 555)
Comments 15: Supplementary materials that are mentioned are not available anywhere for review.
Response 15: We have included the Supplementary Materials in the uploaded files.
Comments 16: Conclusion is missing and needs to be added.
Response 16: We apologize for this error. We’ve added the conclusion section.
Comments 17: How come an IRB exemption was provided for conducting a cross-sectional study? What are the criteria? Additionally, how come a waiver of documentation of consent was used? Is this also determined in the exempt decision?
Response 17: The study was reviewed and approved as exempt because it did not pose more than minimal risk to the participants. The approved exempt protocol granted waiver of documentation of consent and allowed the authors to use an information letter to duly inform patients of what the study was about. All participants received this letter.
Reviewer 2 Report
Comments and Suggestions for Authors
Thank you for sharing this interesting article, “EMSIG: Uncovering Factors Influencing COVID-19 Vaccination Across Different Subgroups Charactered by Embedding-based Spatial Information Gain”. Here are some suggested edits and comments that could help to improve the article:
Major revision
Ø The abstract is overly general and does not provide sufficient detail regarding the study's key findings. It should include the research question, methods, key results, and conclusion. It currently lacks specifics about the methodology and the main outcomes of the research. Add quantitative results to strengthen the abstract.
Ø Revise and summarize the abstract in 250 words.
Ø The full manuscript needs to be revised to remove grammatical errors, such as first mentioning the complete form and then abbreviation for
“COVID-19, and USA”, etc.,
Ø The introduction does not clearly define the research problem.
Ø Expand the background discussion to explain the gap in the literature and the significance of your research in filling that gap.
Ø It is important to state the specific objective of the study early in the introduction to give readers a clear sense of direction.
Ø Full stop after the citation and someplace before the citation. Please use the same pattern.
Ø Most of the cited references are from 2019-2021. Please also cite some recent work from 2021 onwards.
Ø Sections such as the introduction, results, and discussion content can be enriched with research articles.
DOI: 10.3390/vaccines10101583, DOI: 10.3390/ijerph19137926, etc..
Ø Include more recent and relevant citations, especially from the last 5 years, to give a comprehensive overview of the current state of knowledge in the field. This will strengthen the justification for your study.
Ø Write at least one paragraph about the general introduction of COVID-19.
Ø Explain the reason for starting at 18 years of age.
Ø The rationality of the study is not clear.
Ø What is the duration of the study (Date and time)?
Ø The language of the questionnaire?
Ø Informed Consent
Ø How participants were selected for the survey is unclear.
Ø The statistical methods employed are unclear. Please provide detailed information on how data was analyzed and which statistical tests were used. If the sample size is small, justify how the statistical power is sufficient to draw meaningful conclusions.
Ø The results section lacks depth in data interpretation. Several key findings are presented, but the rationale behind their significance is not clearly stated. It is important to better contextualize the results within the framework of the study's objectives.
Ø Figures and tables are difficult to interpret without better labeling and clearer descriptions. For example, Figure 3 appears unclear, and the legend does not provide enough detail to understand the figure's content.
Ø Include more data regarding variability and robustness, such as confidence intervals or error bars, which are currently missing from several key results.
Ø The discussion does not adequately connect the study's findings to the existing literature. More work should be done to compare the results with those of previous studies.
Ø Explain how your results agree with or contradict existing knowledge and why.
Ø There are several places where the discussion makes overly strong conclusions that are not fully supported by the data.
Ø The limitations of the study are not discussed in sufficient detail. It is important to acknowledge any potential experimental design or analysis limitations and suggest how future work might address these issues.
Ø Several figures, including Figure 2 and Table 1, are unclear. The legends are not descriptive enough, and some axis labels are missing or unclear. Ensure all figures and tables are labeled correctly and provide all necessary information for the reader to interpret them without referring to the main text.
Ø Consider improving the visual presentation of the figures to make them more intuitive. Use color coding, clearer legends, or additional annotations to help guide the reader through complex data.
Ø Ensure consistency in terminology and tense throughout the paper.
Ø The conclusion section is weak and does not adequately summarize the implications of the findings. Strengthen this section by restating the key takeaways and providing more concrete statements on the impact of the findings and potential future research directions.
Comments on the Quality of English Language
The full manuscript needs to be revised to remove grammatical errors.
Author Response
|
Response to Reviewer 2’s Comments
1. Summary We appreciate the reviewer's comments and suggestions. Additionally, we’ve made a point-by-point response to the concerns raised by the reviewer. Please find the detailed responses below and the corresponding revisions/corrections highlighted/in track changes in the re-submitted files.
|
|
2. Questions for General Evaluation |
Reviewer’s Evaluation |
Response and Revisions |
|
Does the introduction provide sufficient background and include all relevant references? |
Must be improved |
|
|
Is the research design appropriate? |
Can be improved |
|
|
Are the methods adequately described? |
Can be improved |
|
|
Are the results clearly presented? |
Must be improved |
|
|
Are the conclusions supported by the results? |
Must be improved |
|
- Point-by-point response to Comments and Suggestions for Authors
Comments 1: The abstract is overly general and does not provide sufficient detail regarding the study's key findings. It should include the research question, methods, key results, and conclusion. It currently lacks specifics about the methodology and the main outcomes of the research. Add quantitative results to strengthen the abstract.
Response 1: We have revised the abstract to highlight the specifics of the methodology, key findings, and conclusions. Additionally, we have incorporated quantitative results to enhance the clarity and impact of the abstract.
Comments 2: Revise and summarize the abstract in 250 words.
Response 2: We ensure that the abstract is within the 250-word limit.
Comments 3: The full manuscript needs to be revised to remove grammatical errors, such as first mentioning the complete form and then abbreviation for
“COVID-19, and USA”, etc.,
Response 3: We have reviewed the manuscript to ensure the correctness of the grammar.
Comments 4: The introduction does not clearly define the research problem.
Response 4: We’ve added the sentences about the research problem in the introduction section.
Comments 5: Expand the background discussion to explain the gap in the literature and the significance of your research in filling that gap.
Response 5: We’ve expanded the introduction section explaining the gap, highlighting that long-standing beliefs predating the pandemic contribute to distrust in government and reluctance to receive vaccinations. These entrenched beliefs are difficult to change in the short term, making identifying individuals who are more open to vaccination based on their attitudes essential. Therefore, there is an urgent need to develop a new computational model that efficiently addresses existing limitations and targets populations likely to accept vaccines. Additionally, we addressed the lack of strategies targeting individuals who are more open to vaccination based on their attitudes.
Comments 6: It is important to state the specific objective of the study early in the introduction to give readers a clear sense of direction.
Response 6: We’ve added content to improve clarity and specify the objective of the study. (Lines 126-130)
Comments 7: Full stop after the citation and someplace before the citation. Please use the same pattern.
Response 7: We’ve made the revisions to ensure consistency in the citation format.
Comments 8: Most of the cited references are from 2019-2021. Please also cite some recent work from 2021 onwards.
Response 8: We’ve incorporated the most updated reference in our revision.
Comments 9: Sections such as the introduction, results, and discussion content can be enriched with research articles.
DOI: 10.3390/vaccines10101583, DOI: 10.3390/ijerph19137926, etc..
Response 9: We’ve incorporated additional citations including the two recommended.
Comments 10: Include more recent and relevant citations, especially from the last 5 years, to give a comprehensive overview of the current state of knowledge in the field. This will strengthen the justification for your study.
Response 10: We’ve incorporated the most updated reference in our revision.
Comments 11: Write at least one paragraph about the general introduction of COVID-19.
Response 11: We’ve added more content about the general introduction of COVID-19 in the introduction section. (Lines 41-44, 54-64)
Comments 12: Explain the reason for starting at 18 years of age.
Response 12: The reason for why we only requested responses from people, aged 18 or older is purely because, in Alabama, people who are 18 or older can consent to participate in IRB-approved research at accredited colleges and universities.
Comments 13: The rationality of the study is not clear.
Response 13: We’ve expanded the content in the introduction section to provide additional explanation of the study's rationale and significance. (Lines 90-94)
Comments 14: What is the duration of the study (Date and time)?
Response 14: The data collection started from Feb 1st 2024 and ended on March 11th 2024.
Comments 15: The language of the questionnaire?
Response 15: The administered questionnaire was only in the English language.
Comments 16: How participants were selected for the survey is unclear.
Response 16: Participants were recruited from a Qualtrics panel, a survey sampling company that engages a pre-established group of individuals (referred to as a panel) to respond to surveys based on specified criteria. For our study, we required participants to be at least 18 years old, reside in Alabama, and meet quotas based on COVID-19 vaccination status and race.
Comments 17: The statistical methods employed are unclear. Please provide detailed information on how data was analyzed and which statistical tests were used. If the sample size is small, justify how the statistical power is sufficient to draw meaningful conclusions.
Response 17: We’ve revised the manuscript to provide a detailed description of the statistical methods used, including the chi-square test conducted for statistical analysis as well as the strategy to deal with small sample sizes. (Lines 253-260)
Comments 18: The results section lacks depth in data interpretation. Several key findings are presented, but the rationale behind their significance is not clearly stated. It is important to better contextualize the results within the framework of the study's objectives.
Response 18: We’ve revised the results section to provide a more thorough interpretation of the data, highlighting the significance of key findings and their relevance to the study's objectives. We ensure that the results are better contextualized to align with the overall framework of the study.
Comments 19: Figures and tables are difficult to interpret without better labeling and clearer descriptions. For example, Figure 3 appears unclear, and the legend does not provide enough detail to understand the figure's content.
Response 19: We’ve revised the figures and tables to include clearer labeling and provide more detailed descriptions in the legends, especially for Figure 3, to ensure they are easy to interpret and understand.
Comments 20: Include more data regarding variability and robustness, such as confidence intervals or error bars, which are currently missing from several key results.
Response 20: In the primary feature analysis utilizing regression-based supervised learning, we conducted a sensitivity analysis to evaluate the robustness of our results by resampling the data with 10 different seed values. The COVID-19 perceived detriment, fear, and side effects remained in the top 5 list. In the demographic characterization, we conducted chi-square tests to identify significant proportion changes, using a p-value cutoff of 0.05 to ensure the findings are meaningful. Since the variables were categorical rather than numerical, error bars and confidence intervals were not applicable.
Comments 21: The discussion does not adequately connect the study's findings to the existing literature. More work should be done to compare the results with those of previous studies.
Response 21: We’ve expanded our discussion to include a comparison of our results with existing literature. (Lines 478, 493, 501-502, 521-522, 534, and 558)
Comments 22: Explain how your results agree with or contradict existing knowledge and why.
Response 22: We’ve added explanations for the possible reasons for the differences between existing knowledge and our discovery in the revision.
Comments 23: There are several places where the discussion makes overly strong conclusions that are not fully supported by the data.
Response 23: We’ve removed the overly strong conclusions in the revision, particularly regarding the similarities in geospatial locations of the ROI subgroups and the correlation between flu shot uptake and COVID-19 vaccinations.
Comments 24: The limitations of the study are not discussed in sufficient detail. It is important to acknowledge any potential experimental design or analysis limitations and suggest how future work might address these issues.
Response 24: We’ve included a detailed discussion of the study's limitations, expanding the discussion about experimental design or analysis limitations and how we address the issues in future work. (Lines 557-567)
Comments 25: Several figures, including Figure 2 and Table 1, are unclear. The legends are not descriptive enough, and some axis labels are missing or unclear. Ensure all figures and tables are labeled correctly and provide all necessary information for the reader to interpret them without referring to the main text.
Response 25: We’ve revised the figures and tables to include clearer labeling and provide more detailed descriptions in the legends
Comments 26: Consider improving the visual presentation of the figures to make them more intuitive. Use color coding, clearer legends, or additional annotations to help guide the reader through complex data.
Response 26: We’ve improved the visual presentation by adding color coding, color legends, and new annotations to the revision.
Comments 27: Ensure consistency in terminology and tense throughout the paper.
Response 27: We’ve revised the manuscript to ensure consistency in terminology and tense.
Comments 28: The conclusion section is weak and does not adequately summarize the implications of the findings. Strengthen this section by restating the key takeaways and providing more concrete statements on the impact of the findings and potential future research directions.
Response 28: We’ve revised it to summarize the key takeaways from our findings, emphasizing their implications for inferring tailored interventions and characterizing the subgroups by demographic features. Additionally, we have included specific statements regarding future research directions, particularly focusing on strategies to address vaccine hesitancy. (Lines 572-585)
Comments on the Quality of English Language
Comments 29: The full manuscript needs to be revised to remove grammatical errors.
Response 29: We’ve thoroughly reviewed the manuscript to ensure that it is free of grammatical errors.
Reviewer 3 Report
Comments and Suggestions for Authors
Dear authors,
congratulations on the paper, it is very interesting, but it needs some improvements before being published. Below are my comments:
- if the authors all have the same affiliations I recommend writing the affiliation once.
- In the title I suggest removing EMSIG.
- I suggest dividing better the abstract: introduction, materials and methods, results and conclusions.
- Line 49-79: To improving this part I suggest the following article: Guarducci G, Mereu G, Golinelli D, Galletti G, Gemmi F, Cartocci A, Holczer N, Bacci L, Sergi A, Messina G, Mari V, Nante N. Factors Influencing the Healthcare Workers' Willingness to Receive the COVID-19 Booster Dose in Tuscany (Italy). Vaccines (Basel). 2023 Nov 24;11(12):1751. doi: 10.3390/vaccines11121751.
- I recommend defining the objectives more briefly and concisely.
- I suggest including the questionnaire in the supplementary materials.
- Define whether there were any excluded questionnaires, if so whether a missing data analysis was done. Also, clarify how many people were sent the questionnaire so as to assess the percentage of responders.
- Line 436: I suggest writing "physician" not "doctor".
- Line 444: Please remove the link and add it as references.
- The authors should create a dedicated paragraph for conclusions.
Author Response
|
Response to Reviewer 3’s Comments
1. Summary We appreciate the reviewer's comments and suggestions. Additionally, we’ve made a point-by-point response to the concerns raised by the reviewer. Please find the detailed responses below and the corresponding revisions/corrections highlighted/in track changes in the re-submitted files.
|
|
2. Questions for General Evaluation |
Reviewer’s Evaluation |
Response and Revisions |
|
Does the introduction provide sufficient background and include all relevant references? |
Must be improved |
|
|
Is the research design appropriate? |
Can be improved |
|
|
Are the methods adequately described? |
Must be improved |
|
|
Are the results clearly presented? |
Yes |
|
|
Are the conclusions supported by the results? |
Must be improved |
|
- Point-by-point response to Comments and Suggestions for Authors
Comments 1: if the authors all have the same affiliations I recommend writing the affiliation once.
Response 1: Change has been made as suggested.
Comments 2: In the title I suggest removing EMSIG.
Response 2: We prefer to retain EMSIG as the featured method developed in this study, as it highlights the novel approach and central focus of our work.
Comments 3: I suggest dividing better the abstract: introduction, materials and methods, results and conclusions.
Response 3: We’ve revised the abstract to divide it into subsections to improve the structure.
Comments 4: Line 49-79: To improving this part I suggest the following article: Guarducci G, Mereu G, Golinelli D, Galletti G, Gemmi F, Cartocci A, Holczer N, Bacci L, Sergi A, Messina G, Mari V, Nante N. Factors Influencing the Healthcare Workers' Willingness to Receive the COVID-19 Booster Dose in Tuscany (Italy). Vaccines (Basel). 2023 Nov 24;11(12):1751. doi: 10.3390/vaccines11121751.
Response 4: We’ve revised the introduction section based on the structure of the recommended paper.
Comments 5: I recommend defining the objectives more briefly and concisely.
Response 5: We’ve added the three objectives in the introduction section.
Comments 6: I suggest including the questionnaire in the supplementary materials.
Response 6: We’ve added the questionnaire in Supplementary File 1.
Comments 7: Define whether there were any excluded questionnaires, if so whether a missing data analysis was done. Also, clarify how many people were sent the questionnaire so as to assess the percentage of responders.
Response 7: In the 85-item questionnaire, no missing data were found for any responses after the quality check. However, for the question regarding chronic conditions, participants with missing responses were retained in the analysis. We had a contract with Qualtrics to collect a set quota of 1,000 responses. Qualtrics was responsible for recruiting participants and ensuring that the sample met our inclusion criteria and was representative of Alabama’s population in terms of race, ethnicity, and vaccination status. Qualtrics sent out 3,951 invitations and had 3,101 survey entrants. After excluding 2,081 responses due to various reasons including quality terminations, in-survey terminations, overquota terminations, and data quality checks, 1,020 valid and complete responses remained.
Comments 8: Line 436: I suggest writing "physician" not "doctor".
Response 8: We've made revisions to improve the accuracy of the terminology. Since the original scale refers to 'doctor,' we cannot change this term as it's directly tied to the data. The only instance where we used 'physician' is in the discussion section, when citing a reference paper. (Lines 515 and 517)
Comments 9: Line 444: Please remove the link and add it as references.
Response 9: We’ve revised it.
Comments 10: The authors should create a dedicated paragraph for conclusions.
Response 10: We’ve added the conclusion section.
Reviewer 4 Report
Comments and Suggestions for Authors
Thank you for the opportunity to review this manuscript. This paper introduces the Embedding-based Spatial Information Gain (EMSIG) method to analyze COVID-19 vaccine hesitancy in Alabama by identifying subgroups with distinct perceptions about vaccination. The article presents many strengths and some weaknesses to address. I will proceed with the observations by section below.
Introduction: the paper estalishes the contextual relevance clearly and the introduction outlines the objective to develop the EMSIG method for identifying subgroup-specific concerns. One observation about this section is the geographic focus: while the focus on Alabama is relevant, the introduction does not address whether the findings could be applicable to other regions with similar hesitancy issues.
Methods: the strenghts of the methodology include the use of a structured questionnaire targeting a demographically diverse sample representative of Alabama, considering factors such as race, ethnicity, and political affiliation, and the use of validated scales (CoVaH, T-DiG, TRUST-Ph, TiPHA) as well. Furthermore, the methodology presented (EMSIG) is innovative and demonstrates a forward-thinking approach. However, though the methodology is thorough, it does not address potential bias in survey responses. For instance, participants' self-reporting of vaccine uptake and hesitancy could be influenced by recall bias or social desirability bias, which might distort the data.
Results: the 16 factors extracted from the the 85-item survey across the four scales provide comprehensive insights into vaccine perceptions. The identification of four ROI subgroups with significant variation in vaccine doses and demographics (e.g., age, race, political affiliation) is a key contribution. Also, the random forest regressor model achieves an R² value of 0.446, suggesting moderate predictive success for vaccine uptake based on the 16 factors. However, the results do not account for when participants received their vaccinations, a factor that could significantly influence hesitancy. Participants might have become more or less hesitant over time, but the cross-sectional nature of the study cannot capture these temporal changes. Also, the paper does not address vaccination status related to prior COVID-19 infections. Individuals who may have avoided vaccination due to natural immunity from infection are not distinguished from those who are hesitant for other reasons.
Discussion: the discussion proposes practical interventions based on EMSIG’s findings and rightly highlights the importance of subgroup analysis in understanding vaccine hesitancy. Two weaknesses that could be addressed are: a) a discussion of how EMSIG could be customized for broader applications, e.g. for other regions or national-level datasets; and, b) considerations about whether political or demographic differences between regions (e.g., urban vs. rural, North vs. South) might affect the applicability of the findings.
One final suggestion is adding a reflection about future studies that could consider the impact of different vaccine types (e.g., Moderna, Pfizer, Johnson & Johnson) on hesitancy: different vaccines have varying public perceptions, side effects, and efficacies, which may influence participant responses and willingness to vaccinate.
Author Response
|
Response to Reviewer 4’s Comments
1. Summary We appreciate the reviewer's comments and suggestions. Additionally, we’ve made a point-by-point response to the concerns raised by the reviewer. Please find the detailed responses below and the corresponding revisions/corrections highlighted/in track changes in the re-submitted files.
|
|
2. Questions for General Evaluation |
Reviewer’s Evaluation |
Response and Revisions |
|
Does the introduction provide sufficient background and include all relevant references? |
Yes |
|
|
Is the research design appropriate? |
Can be improved |
|
|
Are the methods adequately described? |
Can be improved |
|
|
Are the results clearly presented? |
Can be improved |
|
|
Are the conclusions supported by the results? |
Can be improved |
|
- Point-by-point response to Comments and Suggestions for Authors
Comments 1: Introduction: the paper establishes the contextual relevance clearly and the introduction outlines the objective to develop the EMSIG method for identifying subgroup-specific concerns. One observation about this section is the geographic focus: while the focus on Alabama is relevant, the introduction does not address whether the findings could be applicable to other regions with similar hesitancy issues.
Response 1: We’ve incorporated the sentences highlighting the aspect that Alabama as a representative politically conserve state in the Deep South, serves as a valuable case study for examining vaccine hesitancy in the introduction section.
Comments 2: Methods: the strenghts of the methodology include the use of a structured questionnaire targeting a demographically diverse sample representative of Alabama, considering factors such as race, ethnicity, and political affiliation, and the use of validated scales (CoVaH, T-DiG, TRUST-Ph, TiPHA) as well. Furthermore, the methodology presented (EMSIG) is innovative and demonstrates a forward-thinking approach. However, though the methodology is thorough, it does not address potential bias in survey responses. For instance, participants' self-reporting of vaccine uptake and hesitancy could be influenced by recall bias or social desirability bias, which might distort the data.
Response 2: We agree that EMSIG does not address potential biases, such as recall or social desirability bias, in the survey. This limitation is acknowledged and discussed in detail in the discussion section.
Comments 3: Results: the 16 factors extracted from the 85-item survey across the four scales provide comprehensive insights into vaccine perceptions. The identification of four ROI subgroups with significant variation in vaccine doses and demographics (e.g., age, race, political affiliation) is a key contribution. Also, the random forest regressor model achieves an R² value of 0.446, suggesting moderate predictive success for vaccine uptake based on the 16 factors. However, the results do not account for when participants received their vaccinations, a factor that could significantly influence hesitancy. Participants might have become more or less hesitant over time, but the cross-sectional nature of the study cannot capture these temporal changes. Also, the paper does not address vaccination status related to prior COVID-19 infections. Individuals who may have avoided vaccination due to natural immunity from infection are not distinguished from those who are hesitant for other reasons.
Response 3: We agree that the results do not account for the timing of when participants received their vaccinations, which may reduce the model's accuracy without this temporal data. Additionally, we acknowledge the limitation of excluding participants who did not receive the vaccination due to a prior COVID-19 infection. These points are addressed in the discussion section.
Comments 4: Discussion: the discussion proposes practical interventions based on EMSIG’s findings and rightly highlights the importance of subgroup analysis in understanding vaccine hesitancy. Two weaknesses that could be addressed are: a) a discussion of how EMSIG could be customized for broader applications, e.g. for other regions or national-level datasets; and, b) considerations about whether political or demographic differences between regions (e.g., urban vs. rural, North vs. South) might affect the applicability of the findings.
Response 4: We’ve incorporated the two points and expanded them into the fourth point of the limitations within the discussion section.
Comments 5: One final suggestion is adding a reflection about future studies that could consider the impact of different vaccine types (e.g., Moderna, Pfizer, Johnson & Johnson) on hesitancy: different vaccines have varying public perceptions, side effects, and efficacies, which may influence participant responses and willingness to vaccinate.
Response 5: We’ve expanded discussion about the type of the vaccine suggested into the first point of the limitation within the discussion section.
Round 2
Reviewer 1 Report
Comments and Suggestions for Authors
I would like to thank the Authors for revising their manuscript and addressing most of my comments.
Remaining comment: - Response regarding the response rate is not included in the text of the paper.
Author Response
- Summary
Thank you very much for your thoughtful feedback. We greatly appreciate your insights and suggestions, which were instrumental in refining our work. Please find the detailed responses below and the corresponding revisions/corrections highlighted/in track changes in the re-submitted files.
|
2. Questions for General Evaluation |
Reviewer’s Evaluation |
Response and Revisions |
|
Does the introduction provide sufficient background and include all relevant references? |
Yes |
|
|
Is the research design appropriate? |
Yes |
|
|
Are the methods adequately described? |
Yes |
|
|
Are the results clearly presented? |
Yes |
|
|
Are the conclusions supported by the results? |
Yes |
|
|
3. Point-by-point response to Comments and Suggestions for Authors |
Comments 1: Response regarding the response rate is not included in the text of the paper.
Response 1: Thanks. We’ve updated the Methods section to include the steps taken by Qualtrics for processing and the response rate. (Lines 145-148)
Reviewer 2 Report
Comments and Suggestions for Authors
Dear Authors,
Thank you for your detailed responses to the comments and suggestions from the initial review. We appreciate the effort and care you have taken to address the concerns raised. After reviewing the revisions made in the resubmitted manuscript, I am pleased to inform you that the revised paper is now acceptable for publication.
Thank you once again for your work on this manuscript.
Author Response
We truly appreciate the constructive feedback provided during the review process, which helped us improve the quality and clarity of our work. We are thrilled to hear that the revised paper is now accepted for publication.
Reviewer 3 Report
Comments and Suggestions for Authors
Dear Authors
Thank you for your effort and the clear answers you gave me. I only have two things that I ask you to add:
- I don't see the recommended article in the bibliography, although we have been treated to the topic. Please if it is as you say in the comments include it.
-In the supplementary I don't see the questionnaire being included, only pictures of the analysis.
Author Response
|
1. Summary Thank you very much for your feedback. We’ve made a point-by-point response to the concerns raised by the reviewer. Please find the detailed responses below and the corresponding revisions/corrections highlighted/in track changes in the re-submitted files.
|
|
2. Questions for General Evaluation |
Reviewer’s Evaluation |
Response and Revisions |
|
Does the introduction provide sufficient background and include all relevant references? |
Can be improved |
|
|
Is the research design appropriate? |
Can be improved |
|
|
Are the methods adequately described? |
Can be improved |
|
|
Are the results clearly presented? |
Yes |
|
|
Are the conclusions supported by the results? |
Yes |
|
- Point-by-point response to Comments and Suggestions for Authors
Comments 1: I don't see the recommended article in the bibliography, although we have been treated to the topic. Please if it is as you say in the comments include it.
Response 1: Apologies for the confusion. In response to the previous Comment 4, we revised the introduction to follow the structure of the recommended paper, which specifically focuses on COVID-19 vaccine adherence among healthcare workers. To address your feedback, we have now included the recommended paper as a citation in the bibliography, and it is also referenced in the updated introduction at Line 63.
Comments 2: In the supplementary I don't see the questionnaire being included, only pictures of the analysis.
Response 2: We’ve included Supplemental File 1 as a separate document in the uploaded materials. Now, we have concatenated all supplementary files, ensuring they are fully accessible to the reviewers.